# On the Application of K-User MIMO for 6G Enhanced Mobile Broadband [note 1]

**DOI:** 10.3390/s20216252

**Published:** 2020-11-02

**Authors:** Anil Kumar Yerrapragada, Brian Kelley

**Affiliations:** Department of Electrical and Computer Engineering, University of Texas at San Antonio, San Antonio, TX 78249, USA; dr.brian.kelley@gmail.com

**Keywords:** beyond-5G, 6G, MIMO, interference alignment, K-User MIMO, OFDM

## Abstract

This paper presents a high-throughput wireless access framework for future 6G networks. This framework, known as K-User MIMO, facilitates all-to-all communication between K access points and K mobile devices. For such a network, we illustrate the demodulation of $K^{2}$ independent data streams through a new interference cancellation beamforming algorithm that improves spectral efficiency compared to massive MIMO. The paper derives a multi-user Shannon Capacity formula for K-User MIMO when K is greater than or equal to 3. We define an Orthogonal Frequency Division Multiplexing (OFDM) frame structure that demonstrates the allocation of time-frequency resources to pilot signals for channel estimation. The capacity formula is then refined to include realistic pilot overheads. We determine a practical upper-bound for MIMO array sizes that balances estimation overhead and throughput. Lastly, simulation results show the practical capacity in small cell geometries under Rayleigh Fading conditions, with both perfect and realistic channel estimation.

## 1. Introduction

The goal of 5G is to enable a fully connected society such that instant information is available just a touch away. 5G achieves this through three key paradigms viz., enhanced Mobile Broadband (eMBB) for gigabit data rates, Ultra Reliable and Low Latency Communications (URLLC) for latency less than 1 ms, and massive Machine-Type Communications (mMTC) for 1millionconnecteddevices/sq.km [1,2]. These paradigms are supported by a multilayer technology strategy including small cell architectures [3,4], millimeter wave communication, and massive MIMO. Millimeter wave systems facilitate communication in the high Radio Frequency (RF) bands using analog, digital, and hybrid beamforming [5,6]. Massive MIMO deploys large antenna arrays at base stations and operates in the low to mid RF bands [7,8,9,10]. To support the ubiquitous deployment of densely connected networks, this paper investigates an alternative MIMO technology, in the microwave realm, for beyond-5G or 6G networks. This technology, known as *K*-User MIMO, has the potential to achieve very high throughput compared to 5G Massive MIMO.

### Background and Prior Research on K-User MIMO

*K*-User MIMO is an architecture in which there are *K* access points and *K* mobile devices, each equipped with multiple antennas, i.e., spatial dimensions. In the simplest form of *K*-User MIMO, each Access Point connects to one of the *K* mobile devices. In this paper, we consider a different form of *K*-User MIMO, known as *K*-User MIMO X, in which each of the *K* mobile devices receives signals from *K* access points. This is shown in Figure 1. The all-to-all architecture achieves very high throughput whilst supporting flexibility in achieving diversity. Each of the *K* access points could send redundant information streams to maximize reliability. Alternatively, they could send unique information streams in order to maximize capacity. *K*-User MIMO X can switch between these two modes without any change in the mathematics of the algorithm. Further scenarios can be envisioned in which *K*-User MIMO X allows for adaptive allocation of power to users with more favorable Signal-to-Noise Ratios. Additional encoding of data across time and frequency could be applied so the signals could adapt to malicious behavior such as jamming and eavesdropping.

In any form of *K*-User MIMO, each mobile device receives both its desired signals and interference signals (signals meant for other mobile devices). To manage interference, *K*-User MIMO systems are often studied in the context of Interference Alignment (IA). IA is a technique that aligns interfering signal vectors in order to maximize interference-free space at each mobile device [11]. By applying suitable channel dependent precoders to the transmit signals, and beamformers to the receive signals, several interfering users can communicate simultaneously. Alignment helps confine the interference at each mobile device to a smaller dimensional subspace while projecting the desired signals into the null space of the interference.

Several works have analyzed IA on a theoretical level. A typical metric used to characterize IA is known as Degrees of Freedom (DoF). DoF is defined as the number of spatial dimensions that are free from interference [12]. The authors in [13] have provided examples showing acheivability of IA and various DoF in *K*-User interference networks with different antenna configurations. In [14], an iterative algorithm for obtaining the precoders and beamformers is presented for a Time Division Duplex (TDD) mode of operation. The precoders in this method are a function of the dual relationship between the MIMO forward and reverse channels. Another IA framework involving TDD channels is presented in [15]. In [16], interference alignment in MIMO downlink networks is investigated, where precoders are derived by eigen decomposition of the MIMO channels.

Another IA scheme for a *K*-User MIMO X network is proposed in [17]. By appropriately precoding the transmit signals, this scheme maximizes the interference-free space by limiting the interference at every mobile device to half of the received signal space. Further, by applying a zero forcing beamformer which is a function of interfering channels and precoders, interference cancellation has been achieved for K=3. The algorithms in [17] are purely theoretical and in this paper, we improve upon them.

Discussion on how to demodulate symbols is not provided in [17]. Further, in order to maximize capacity, we wish to operate the *K*-User MIMO X system such that each access point is transmitting different symbols to each mobile device, whilst all being on the same frequency subcarrier. We have investigated the case where each access point is on a different subcarrier in [18] and found that the bandwidth and therefore the capacity of the *K*-User system is reduced by a factor of *K*, which is significant. This paper presents a new signal separation beamformer to regain the lost factor of *K*.

Interference Alignment uses precoders and beamformers that are channel dependent. Naturally, channel estimation is critical to IA [19,20]. Both [17] and our previous works [18,21,22] either consider perfect channel state information or do not consider exact channel estimation error models. Neither considers the overheads arising from transmitting pilot symbols for estimation. Park and Ko [17] assumes perfect channel state information. Yerrapragada and Kelley [18] does not explicitly estimate the channel but assumes a Cramér–Rao variance for the estimation error, which is only a lower-bound on the error. In [21], the channel is not estimated but the effects of imperfect estimation are simulated. Residual interference due to imperfect precoders and beamformers is modeled as a random variable and an expression for its distribution is provided. Yerrapragada and Kelley [22] is the first paper that introduces signal separation concepts for *K*-User MIMO, but it too assumes perfect channel estimation.

In this paper, we extend our previous work to provide realistic and practical capacity results for *K*-User MIMO X systems that account for channel estimation overheads.

## 2. System Model for Cellular-Based K-User MIMO

This section describes the system model for *K*-User MIMO X that shows transmit precoding, receive interference cancellation, and signal separation. Table 1 shows the parameters used in the key equations and derivations. Figure 2 shows the overall *K*-User MIMO protocol steps. Lastly, an Orthogonal Frequency Division Multiplexing (OFDM) multiple access protocol is shown. This protocol illustrates the pilot overhead resulting from serving several (>K) mobile devices.

### 2.1. Received Signal at Antenna

Each of the *K* access points and mobile devices is equipped with *M* antennas. The all-to-all connectivity results in each mobile device receiving *K* desired signals and K(K−1) interfering signals. This is shown in Figure 2 and Equation (Equation 1). Transmitted symbols $s_{ij}$ between the $j^{th}$ access point and the $i^{th}$ mobile device are precoded by length *M* precoder vectors $v_{ij}$ and transmitted over Rayleigh fading channels $H_{ij}$. Without loss of generality, we consider downlink transmissions. The received signal at the $i^{th}$ mobile device is given by
(1)$$y_{i}=\underset{\mathrm{Desired}\;\mathrm{signal}}{\underset{︸}{\sum_{j=1}^{K}H_{ij}v_{ij}s_{ij}}}+\underset{\mathrm{Total}\;\mathrm{interference}}{\underset{︸}{\sum_{j=1}^{K}\sum_{k=1,\neq i}^{K}H_{ij}v_{kj}s_{kj}}}+\underset{\mathrm{Noise}}{\underset{︸}{w_{i}}}.$$

This paper focuses on the maximum capacity scenario. Therefore it is assumed that each $s_{ij}$ is unique. The precoders $v_{ij}$ are channel dependent and are obtained by solving a system of alignment equations. The procedure for obtaining the precoder vectors is shown in Appendix A. The noise at the $i^{th}$ mobile device, $w_{i}$ is assumed to be 0 mean Additive White Gaussian Noise (AWGN) with variance $\sigma_{w_{i}}^{2}=E\left[w_{i}w_{i}^{*}\right]$, where E[.] represents the expected value.

### 2.2. Received Signal after Beamformer for Interference Cancellation

A beamformer matrix $U_{i}$ is applied to the received signal $y_{i}$ to cancel the interference, as shown below: (2)$$y_{i}^{1}=U_{i}^{H}\cdot y_{i}=U_{i}^{H}\sum_{j=1}^{K}H_{ij}v_{ij}s_{ij}+\underset{I_{\epsilon }\approx 0}{\underset{︸}{U_{i}^{H}\cdot \sum_{j=1}^{K}\sum_{k=1,\neq i}^{K}H_{ij}v_{kj}s_{kj}}}+U_{i}^{H}w_{i},$$
where $I_{\epsilon }$ is the residual interference after cancellation. In the case of perfect channel state information, it is exactly equal to zero. The interference cancellation beamformer $U_{i}$ is obtained by first generating a matrix, whose columns contain aligned interference vectors. The left-hand side of the Singular Value Decomposition (SVD) of this matrix contains the beamformer. The derivation of this beamformer is shown in Appendix A.

### 2.3. Received Signal after Beamformer for Desired Signals Separation

Symbols transmitted from access points on the same frequency subcarrier, add coherently at the mobile device. Under this model, symbol recovery is straightforward when each access point sends the same symbol to the $i^{th}$ mobile device. However, for maximizing capacity, each access point must be able to send different symbols to the $i^{th}$ mobile device. In such cases, a second beamformer operator is applied after interference cancellation, as shown in Equation (Equation 3). This second operator is applied *K* times at each mobile device (shown in Figure 1) and helps separate the signals sent from each of the *K* access points.
(3)$$\begin{array}{cc} y_{ij}^{2}=U_{ij}^{H}U_{i}^{H}y_{i}^{1} & =U_{ij}^{H}U_{i}^{H}\sum_{j=1}^{K}H_{ij}v_{ij}s_{ij}+\underset{\approx 0}{\underset{︸}{U_{ij}^{H}I_{\epsilon }}}+U_{ij}^{H}U_{i}^{H}w_{i}\\ \; & =U_{ij}^{H}U_{i}^{H}H_{ij}v_{ij}s_{ij}+U_{ij}^{H}U_{i}^{H}w_{i}\overset{\mathrm{To}\;\mathrm{detector}\;\mathrm{for}\;\mathrm{symbol}\;\mathrm{recovery}}{\to }{\hat{s}}_{ij}. \end{array}$$

The signal separation beamformer $U_{ij}$ is obtained by first generating a matrix whose columns contain desired signal vectors from other access points that act as interference when recovering the symbol transmitted from a specific access point. The left-hand side of the SVD of this matrix contains the beamformer. The derivation of this beamformer is shown in Appendix A.

After separation, the desired signals from each of the *K* access points can be decoded by correcting for the effects of $U_{ij}$, $U_{i}$, $H_{ij}$, and $v_{ij}$. The decoding and detection process that recovers the symbols $s_{ij}$ is shown in Appendix A.

### 2.4. K-User MIMO X Multiple Access Protocol

We now apply *K*-User MIMO X to a typical OFDM cellular scenario. Let us assume that there are nK mobile devices (n≥1) associating with *K* access points forming *nK*-User-Groups (UG). In a single UG, there are $K^{2}M^{2}$ Channel Impulse Responses (CIR) that need to be estimated. We propose that each of the MK transmit antennas sends pilot symbols on non-overlapping OFDM symbol times. In this paper, we assume the use of pilot signals that are configured for both synchronization as well as channel estimation. An example of such a sequence is the Zadoff–Chu sequence, commonly used in 4G Long Term Evolution (LTE). Akin to LTE pilot signals, when one transmit antenna is sending pilots, all other transmit antennas are off. This is shown in Figure 3. In the time domain, the channel needs to remain constant for at least MK symbol times. We leverage frequency domain resources to support the *n* User Groups. In the frequency domain, the available bandwidth *B* is divided into *m* sub-bands, where $m=\frac{B}{B_{c}}$ and $B_{c}$ is the channel coherence bandwidth. Each of the *m* sub-bands is divided equally among the *nK*-User-Groups. As shown in Figure 3, each UG gets a chunk of bandwidth in each of the *m* sub-bands in which to transmit pilot signals for synchronization and channel estimation. The channels in the sub-bands not available for a certain UG can easily be obtained by interpolation. The channel estimates are conveyed by the mobile devices to a global network entity in the backhaul which makes all channels available to all access points and mobile devices through the appropriate interfaces. The overhead from this step is not considered in this paper and will make up future work.

## 3. Derivation of Shannon Capacity for K-User MIMO X and Small Cell Geometric Capacity in Rayleigh Fading

This section presents an analysis of Shannon Capacity as a function of *K*, with and without pilot overheads. Further simulation results show the statistical distributions of capacity for K=3 in Rayleigh fading and small cell geometries.

### 3.1. Ideal K-User MIMO X Capacity vs. K Excluding Pilot Overhead

The theorem for the upper-bound capacity for *K*-User MIMO X incorporates channel and beamformer gains that scale with *K* and antenna array size *M*. It is defined below.

**Theorem** **1.***The upper bound multi-user capacity of a K-User MIMO X system is bounded by $C_{bits/sec}\leq BK^{2}log_{2}\left(1+\left[M-\frac{N_{I}}{2}-\left(N_{D}-1\right)\right]{\left[M-\frac{N_{I}}{2}\right]}^{2}M^{2}\times \mathrm{SIN}R\right)$,
where**B**is the bandwidth, and the $SINR=\frac{P_{t}d^{-\alpha }}{\sigma_{w}^{2}}$ includes the transmit power $P_{t}$ and the distance dependence based on a path loss exponent α and target distance**d*. *The proof of Theorem 1 and verification by simulation are shown in Appendix B and Appendix C, respectively. The result of Theorem 1 gives the upper-bound multi-user capacity of the combined $K^{2}$ streams. Note that the theorem represents an unconstrained case which assumes that the entire available time-frequency resources are available only for data transmission.*

### 3.2. Ideal K-User MIMO X Capacity vs. K Including Pilot Overhead

Real systems are impacted by various overheads for synchronization signals, channel estimation pilot signals, and exchange of other control information. In this paper, we consider two of the most important overheads—that of synchronization and channel estimation. We propose the use of pilot signals configured for both time synchronization and channel estimation. Consequently, we define the following theorem that refines Theorem 1 to include pilot overheads.

**Theorem** **2.**
*If MK channel estimation symbols can be transmitted in less than the channel coherence time, i.e., $MKT_{symb}<T_{c}$, the remaining time can be reserved exclusively for data transmission. The capacity equation in Theorem 1 can be modified as follows, $C_{bits/sec}\leq \frac{T_{c}-MKT_{symb}}{T_{c}}BK^{2}log_{2}\left(1+\left[M-\frac{N_{I}}{2}-\left(N_{D}-1\right)\right]{\left[M-\frac{N_{I}}{2}\right]}^{2}M^{2}\times \mathrm{SIN}R\right)$,
where*
$T_{c}$
*is the channel coherence time. $T_{symb}$ is the duration of 1 OFDM symbol given by $\left(N+CP\right)T_{s}$, where *N* is the size of the Fast Fourier Transform (FFT), CP is the size of the Cyclic Prefix in samples, and $T_{s}$ is the sampling period.*


Figure 4 shows the Shannon Capacity curves for *K*-User MIMO X with and without pilot overheads. The key observation from the curves is that while the unconstrained capacity from Theorem 1 continues to grow with *K*, that is not the case when pilot overheads are taken into account. After a certain point, it can be seen that the pilot overheads overwhelm the gains from *K*-User MIMO and the capacity begins to drop. In highly varying channel scenarios, it is more beneficial to operate at a lower value of *K* and schedule several User Groups in a multiple access framework similar to that shown in Figure 3. Figure 4 also compares the performance of our *K*-User MIMO X system with 5G-NR Massive MIMO [23]. It should be noted that the Massive MIMO model assumes one access point equipped with *M* antennas and *K* single-antenna mobile devices. The Shannon limit is also plotted. This limit is based on the analysis provided in [24] and assumes a system with a single access point and single mobile device each with MK antennas.

### 3.3. Capacity Results for Small Cell K-User MIMO at K = 3

This section describes the simulation model, choosing K=3 as an example. Simulations are done in MATLAB and the key parameters are listed in Table 2. Cell spectral efficiency performance in 500 m, 100 m, and 50 m hexagonal cells is obtained and shown in Figure 5.

Without loss of generality, the hexagonal geometry is chosen, for simplicity of analysis. The system can easily be translated to stochastic geometries, commonly associated with 5G systems. 3 access points are placed on alternate corners of the cell. 3 mobile devices are placed uniformly within the cell. The simulations assume an exponential path loss $L_{ij}=d_{ij}^{-\alpha }$.

While *K*-User MIMO X is not limited by any particular channel scenario, we note that one example of a use case is a high-throughput IoT robotic factory model. Hence, without loss of generality, a Rayleigh fading channel model with the Indoor A power delay profile [25] is used. Considering an Indoor A channel scenario with parameters such as velocity v=3kmph, carrier frequency $f_{c}=1.9\;\mathrm{GHz}$, and speed of light $c=3\times 10^{8}\;m/s$, the channel coherence time $T_{c}$ is calculated as $T_{c}=\sqrt{\frac{9}{16\pi f_{d}^{2}}}\approx 80\;\mathrm{ms}$, where $f_{d}$ is the Doppler shift given by $f_{d}=\frac{vf_{c}}{c}$. Let us assume that we only have a fraction of the channel coherence time, say 60ms.

Multiple channel and location trials are run. In each trial, the channel is estimated in the first 18 symbol times (1.5ms). This paper assumes the use of well-known estimation methods such as the Maximum Likelihood (ML) or Minimum Mean Square Error (MMSE).

For statistical analysis, it suffices to assume the presence of one *K*-User-Group. In such a case, the entire bandwidth is available for pilot signals. We use the Zadoff–Chu sequence for both time synchronization and channel estimation. After channel estimation, the remaining 58.5ms is available for data transmission. The Cumulative Distribution Function (CDF) of the spectral efficiency is calculated from the multiple trials and plotted in Figure 5. These spectral efficiencies account for pilot signal overhead as well as estimation error.

In the case of n>1*K*-User-Groups, in the channel estimation symbols, the bandwidth can be divided as shown in Figure 3. In the data transmission symbols, the bandwidth could also be divided into sub-bands in which the different User Groups could be network scheduled using greedy or proportional fair algorithms.

#### Incorporation of Channel Estimation Errors

Figure 6 shows the variance of the estimation error as a function of signal-to-noise ratio for both the ML and MMSE estimates. The Cramér–Rao Lower Bound (CRLB) is also shown. As expected, the ML and MMSE variances are higher than the CRLB. The equations for the channel estimation methods and the CRLB are well researched in other works and are hence shown in Appendix D. Figure 5 also shows the spectral efficiency if the estimation error variance is at the Cramér–Rao Lower Bound (CRLB). To simulate this, we calculate the CRLB as shown in Equation (Equation 18) in Appendix D. The CRLB is calculated for each transmit–receive antenna link, and error terms are drawn from $\mathrm{CN}(0,\;\sigma_{CRLB}^{2})$. These errors are added to the actual channels to simulate estimation error at the CRLB.

It can be seen from Figure 5 that the median best case spectral efficiency in 500 m, 100 m, and 50 m cells are 170, 230, and 256 bits/sec/Hz, respectively. These results are under Rayleigh fading conditions. If the channel follows a Rician distribution, it means that there is one dominant line of sight path in the Channel Impulse Response. The stronger the line-of-sight component, the rarer the occurrences of deep fades. Though this scenario is not simulated explicitly, we have investigated another scenario which is maximum capacity scheduling with Rayleigh fading. This approach schedules *K*-User-Groups in a particular sub-band with the best channel conditions, thus weeding out deep fade channel instances. We have found very little improvement in spectral efficiency with this type of scheduling. We believe that the same will be the case with Rician channels. The reason for this is that the non co-located nature of K-User MIMO transmitters already provides enough channel diversity to overcome the effects of deep fades.

The spectral efficiencies in Figure 5 are multi-user values and take into account the fact that channel estimation through pilot signals takes up 18 symbol times, where no data is transmitted. In a 20 MHz band, this amounts to a best-case scenario of 3.4 Gbps, 4.6 Gbps, and 5.1 Gbps, respectively; and in the case of 5 aggregated bands, data rates in excess of 17 Gbps, 23 Gbps, and 25 Gbps, respectively, can be achieved. This underlines the wide range of exciting possibilities that can be achieved in beyond-5G and 6G networks with *K*-User MIMO X.

The 6G extension of eMBB, defined in [26] as eMBB Plus, will serve mobile and IoT communications with data rate requirements far greater than 5G. The high throughput and spectral efficiency of *K*-User MIMO X lend themselves well to help support eMBB Plus. 6G will also be more machine-learning and security driven [27,28]. The all-to-all nature of *K*-User MIMO X, with its ability to switch between maximum capacity and maximum reliability modes, makes it particularly suitable for future machine learning integration. Machine learning algorithms could be used to come up with new encoding schemes across the *K* transmitters that can help adapt to spatially and temporally varying channel conditions as well as eavesdroppers and jammers. This flexibility leads perfectly into the 6G version of URLLC, known as event-defined-URLLC [28], which provides context-aware communications not thought of in 5G.

## 4. Conclusions

In this paper, we have reviewed and identified the massive scope for increased throughput for beyond-5G or 6G networks. Under realistic channel estimation constraints, we have provided a *K*-User MIMO X framework that can cancel interference, demodulate, and maximize capacity through signal separation. Further practical aspects such as OFDM multiple access for channel estimation and data transmission have been described. Lastly, cell capacity performance has been simulated and compared with related technologies.

## Figures and Tables

**Figure 1 sensors-20-06252-f001:**
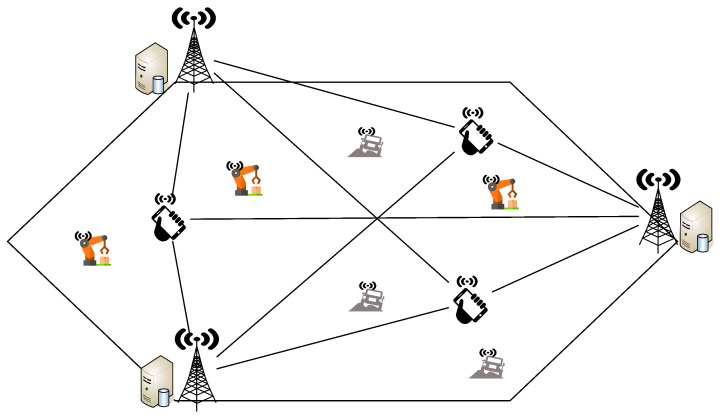
*K*-User MIMO X network Example for K=3.

**Figure 2 sensors-20-06252-f002:**
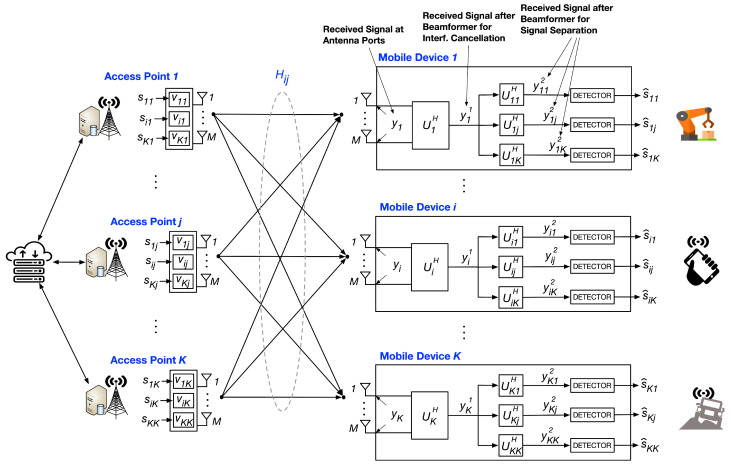
*K*-User MIMO X network showing the application of precoders at the access points as well as stage 1 and stage 2 beamformers (Equations (Equation 1)–(Equation 3)).

**Figure 3 sensors-20-06252-f003:**
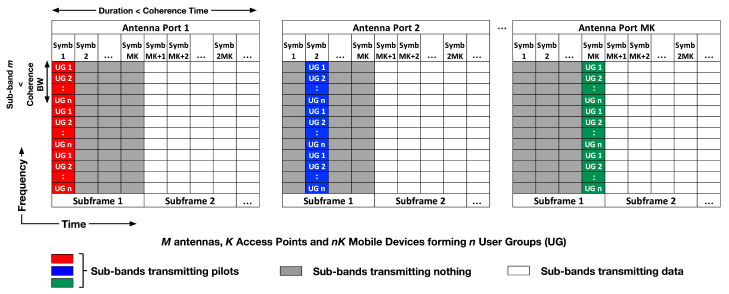
Example showing the channel estimation overhead for *nK*-User-Groups (UG). The bandwidth allocated for estimation is divided among the *nK*-User-Groups. Each group transmits pilot signals in its allocated band.

**Figure 4 sensors-20-06252-f004:**
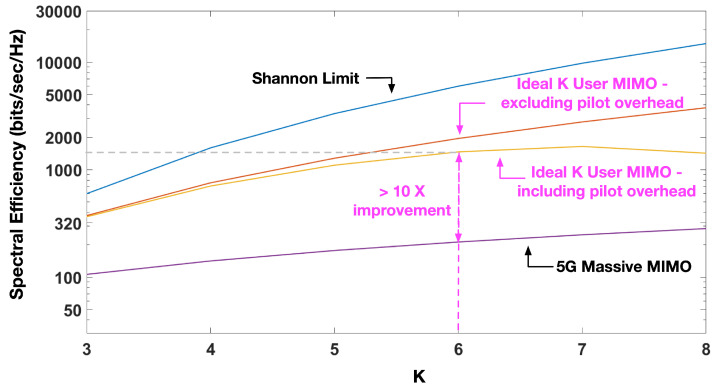
Comparison of our very high throughput *K*-User MIMO X spectral efficiencies against related technologies such as 5G-NR Massive MIMO. Spectral Efficiencies are for a 100 m cell in an Indoor A channel scenario. The curve for Massive MIMO is based on the formula shown in [23]. The number of antennas is M=K(K−1).

**Figure 5 sensors-20-06252-f005:**
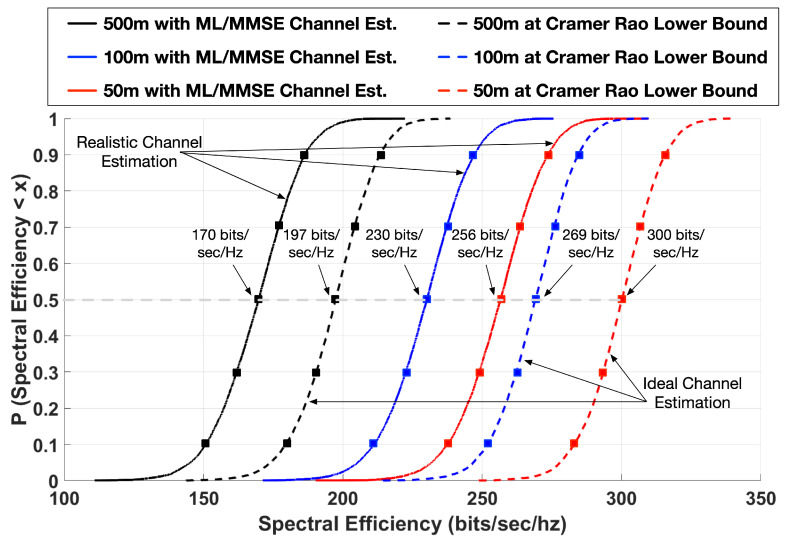
Cumulative distribution functions of spectral efficiencies in bits/sec/Hz for K=3 in single hexagonal cells of radius 50 m, 100 m, and 500 m for Indoor A channel scenarios [25]. Results are shown both with realistic channel estimation using ML/MMSE and also at the Cramér–Rao Lower Bound.

**Figure 6 sensors-20-06252-f006:**
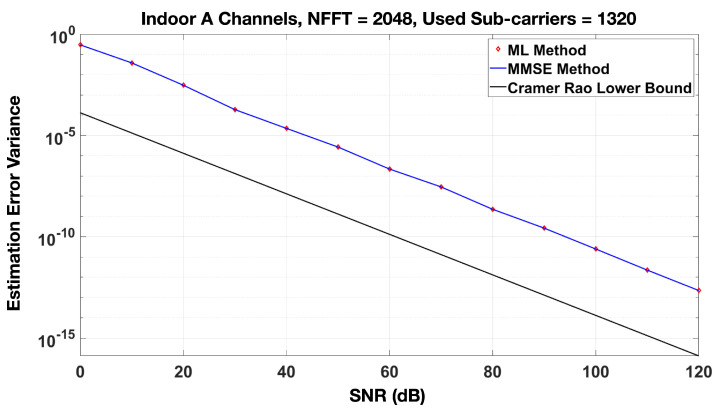
Comparison of channel estimation error variance between Maximum Likelihood (ML)/Minimum Mean Square Error (MMSE) and the Cramér–Rao Lower Bound (CRLB).

**Table 1 sensors-20-06252-t001:** System model parameters for key derivations.

Parameter	Description	Size	For K=3
*M*	Minimum number of antennas at each access point and mobile device	≥K(K−1)	≥6
$N_{D}$	Number of desired signals at each mobile device	*K*	3
$N_{I}$	Number of interference terms at each mobile device	K(K−1)	6
$H_{ij}$	Channel matrix between access point *j* and mobile device *i*	M×M	6×6
$v_{ij}$	Precoder vector for signal between access point *j* and mobile device *i*	M×1	6×1
$s_{ij}$	Symbol to be transmitted between access point *j* and mobile device *i*	1×1	1×1
$w_{i}$	Additive White Gaussian Noise (AWGN) at mobile device *i*	M×1	6×1
$P_{i}^{I}$	Matrix of aligned interference column vectors at the mobile device *i*	$M\times \frac{N_{I}}{2}$	6×3
$U_{i}$	Zero forcing beamformer matrix at mobile device *i*	$M\times M-\frac{N_{I}}{2}$	6×3
$P_{ij}^{D}$	Matrix of desired column vectors at mobile device *i* to isolate signal from access point *j*	$M-\frac{N_{I}}{2}\times N_{D}-1$	3×2
$U_{ij}$	Signal Separation beamformer matrix at mobile device *i* to isolate signal from access point *j*	$M-\frac{N_{I}}{2}\times$ $M-\frac{N_{I}}{2}-\left(N_{D}-1\right)$	3×1
$N_{s}$	Number of recovered copies of each symbol $s_{ij}$	$M-\frac{N_{I}}{2}-\left(N_{D}-1\right)$	1

**Table 2 sensors-20-06252-t002:** Simulation Parameters.

Channel Model	Rayleigh Fading
Channel Scenario	Indoor A [25]
Cell Radius	50 m, 100 m, 500 m
Transmit Power	16 dBW
Total Bandwidth	20 MHz
FFT Size (*N*)	2048
Cyclic Prefix (CP)	512 samples
Sampling frequency	30.72 MHz
Subcarrier spacing	15 kHz
Number of used sub-carriers	1320
Noise Figure	4 dB
Thermal Noise Density	−203.9 dBW/Hz
Path Loss Exponent	3

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
