# Peer review of "On the Application of K-User MIMO for 6G Enhanced Mobile Broadband [Author-notes fn1-sensors-20-06252]"

_sensors, 2020, doi:10.3390/s20216252_

Round 1

Reviewer 1 Report

The authors addressed a K-user MIMO framework for 6G cellular systems. In this paper, they stated the framework description and the mathematical derivations very well. The simulation results are also promising. Overall this paper is very well written and organized.

For further improvement of the paper quality, the reviewer recommends the following tasks:
1. The authors need to clarify how the proposed K-user MIMO framework contributes to 6G cellular systems. The massive MIMO technique supports the 5G scenarios: eMBB, URLLC, and mMTC. What are the 6G scenarios based on the K-user MIMO technique?

2. The proposed K-user MIMO method seems to be an extension of the previous works: [17], [18], [21], and [22]. The authors briefly summarized the methods of [17] and [18]. In addition to [17] and [18], the authors need to describe the approaches of [21] and [22].

Reviewer 2 Report

The paper quality is above average paper level. Very clear introduction shows the paresent state of the investigation on beyond 5G MIMO technology.

The description of research made in the framework of article is presented well, the simulation results are demonstrative and analysed well.

My questions are:

what simulation environment was used for the simulations?

what would be the difference on results if there would be also Rician channel paths (and also Rayleigh channels)?
